# Transcriptome Profiling of the Anterior Cingulate Cortex in a CFA-Induced Inflammatory Pain Model Identifies ECM-Related Genes in a Model of Rheumatoid Arthritis

**DOI:** 10.3390/genes17010015

**Published:** 2025-12-25

**Authors:** Guang-Xin Xie, Jian-Mei Li, Bai-Tong Liu, Jiang-Tao Wang, Lu-Shuang Xie, Xiao-Yi Xiong, Qiao-Feng Wu, Shu-Guang Yu

**Affiliations:** 1College of Basic Medicine, Chengdu University of Traditional Chinese Medicine, Chengdu 611137, China; 2Key Laboratory of Acupuncture for Senile Disease, Ministry of Education, Chengdu University of Traditional Chinese Medicine, Chengdu 610075, China; 3Acupuncture and Chronobiology Key Laboratory of Sichuan Province, Chengdu University of Traditional Chinese Medicine, Chengdu 610075, China; 4Acupuncture and Moxibustion School, Chengdu University of Traditional Chinese Medicine, Chengdu 611137, China

**Keywords:** CFA, whole transcriptome, rheumatoid arthritis, random forest, hub genes

## Abstract

Background: Rheumatoid arthritis (RA) is a chronic autoimmune disease characterized by persistent joint inflammation and progressive bone destruction. However, its complex pathogenesis remains poorly understood, and effective therapeutic targets are still lacking. Objective: This study aimed to identify key genes associated with RA and elucidate their biological significance by integrating bioinformatic analysis with experimental validation. Methods: Whole-transcriptome data from the anterior cingulate cortex (ACC) of Complete Freund’s Adjuvant (CFA)-induced inflammatory pain and control mice (GSE147216 dataset, GEO database) were collected from NCBI (National Center for Biotechnology Information). Differentially expressed genes (DEGs) were first identified. Subsequent analyses included Gene Ontology (GO) and Kyoto Encyclopedia of Genes and Genomes (KEGG) pathway enrichment, construction of a protein–protein interaction (PPI) network, and identification of hub genes using a Random Forest machine learning algorithm. Quantitative PCR (qPCR) was performed to validate gene expression levels. Results: A total of 76 DEGs were identified, including 64 upregulated and 12 downregulated genes. Among them, *Fn1* (fibronectin 1), *Bgn* (biglycan), and *Lum* (lumican) were identified as hub genes. Functional enrichment analysis revealed inflammatory responses, extracellular matrix (ECM) remodeling, and the TGF-β signaling pathway. qPCR validation confirmed significant upregulation of *Fn1*, *Bgn*, and *Lum* mRNA in the CFA group. Conclusions: This study highlights the potential roles of *Fn1*, *Bgn*, and *Lum* in the central sensitization associated with inflammatory pain, offering insights relevant to RA.

## 1. Introduction

RA is a chronic autoimmune disease characterized by synovial inflammation, hyperplasia, and progressive bone destruction, with a global prevalence of approximately 0.5–1% [1]. The disease has a high disability rate and currently lacks curative treatment strategies. The pathogenesis of RA involves aberrant activation of immune cells [2], excessive secretion of proinflammatory cytokines such as TNF-α and IL-6 [3], and the generation of autoantibodies [4], ultimately resulting in joint deformities and loss of function. Although biologic agents such as anti-TNF-α monoclonal antibodies and targeted synthetic disease-modifying antirheumatic drugs (tsDMARDs) have improved outcomes in some patients [5], 30–40% of individuals experience poor therapeutic response or develop drug resistance [6]. Moreover, long-term medication use may cause adverse effects such as infections and hepatic or renal dysfunction. Therefore, it remains essential to further elucidate the molecular regulatory network of RA and identify novel therapeutic targets.

Emerging evidence suggests that RA is not only a peripheral joint disease but also involves complex neuroimmune interactions within the central nervous system (CNS) [7]. For example, the chronic pain in RA is often poorly correlated with peripheral inflammation and joint damage, highlighting the importance of central sensitization [8]. ACC, a key region involved in affective and sensory dimensions, has been increasingly recognized as a central hub in pain modulation in RA [9]. Functional and structural alterations in the ACC have been associated with persistent inflammatory pain and emotional dysregulation in both human and animal studies [10]. Recent preclinical models have demonstrated that neuroinflammatory changes in the ACC contribute to the maintenance of chronic pain, possibly through glial activation, synaptic remodeling, and ECM dysregulation [11]. However, the molecular mechanisms underlying ACC involvement in RA-related pain remain largely unexplored.

The advancement of transcriptomic technologies and systems biology has enabled high-resolution profiling of CNS regions implicated in chronic pain [12]. Bioinformatics tools, including differential expression analysis, PPI network construction, and machine learning algorithms, provide powerful strategies to identify molecular signatures and key regulatory nodes within complex biological systems [13]. The integration of these approaches has been successfully applied to various pain models, yet studies focusing on RA-induced central changes, especially in the ACC, are still limited. Animal models such as CFA-induced inflammatory pain offer a reliable platform to investigate these mechanisms, as they mimic both peripheral inflammation and CNS sensitization components of RA [14]. Specifically, the CFA-induced arthritis model is well-established for studying RA-like pathology, as it recapitulates key features such as synovitis, pain hypersensitivity, and systemic inflammation [15]. In light of this, we employed a transcriptomic dataset (GSE147216) derived from the ACC region of CFA-treated mice to explore key genes and pathways potentially involved in RA. We integrated differential gene expression analysis, functional enrichment, network-based prioritization, and quantitative validation to identify and verify potential central targets in the RA-related inflammatory pain model.

To specifically investigate these central neuroimmune mechanisms, we therefore utilized a CFA-induced unilateral inflammatory pain model [16]. It is important to note that while this model does not replicate the systemic autoimmunity of RA, it robustly mimics key aspects relevant to our study: sustained peripheral inflammation and the subsequent development of central sensitization, particularly within pain-processing brain regions like the ACC [17]. Thus, it serves as a valuable tool for investigating central mechanisms that may also be operative in RA.

## 2. Materials and Methods

### 2.1. Data Acquisition and Processing

The transcriptomic dataset GSE147216 was downloaded from the GEO database (https://www.ncbi.nlm.nih.gov/geo (accessed on 15 October 2025)) [18], which contains whole-transcriptome data from the ACC of six C57BL/6 mice, including three from the CFA model group and three from the control group. Raw data were preprocessed using R software (version 4.0.3), including background correction and normalization. Differential gene expression analysis was performed using the DESeq2 package (version 1.46.0). The resulting *p*-values were adjusted for multiple testing using the Benjamini–Hochberg false discovery rate (FDR) correction to control the false discovery rate at 5% (FDR < 0.05). DEGs were defined as genes with adjusted *p*-value (FDR) < 0.05 and |log2(Fold Change) | ≥ 0.585 [19].

### 2.2. Experimental Animals

Male C57BL/6 mice (22–25 g) were purchased from Chengdu Dossy Experimental Animals Co., Ltd. (Chengdu, China). All animals were housed in a specific pathogen-free (SPF) animal facility under standard laboratory conditions, including a constant temperature of 22–25 °C, humidity of 50–60%, and a 12 h light/dark cycle (lights on from 07:00 to 19:00). Mice had free access to standard laboratory chow and autoclaved tap water. Cages were cleaned regularly, and animals were monitored daily for general health and signs of distress. All experimental procedures were conducted in accordance with animal welfare regulations and designed to minimize pain and discomfort.

### 2.3. GO and KEGG Functional Enrichment Analysis

Functional enrichment analysis of DEGs was conducted using the ClusterProfiler package in R (version 4.14.6), focusing on GO and KEGG pathways [20]. GO analysis was categorized into three domains: biological process, cellular component, and molecular function. The enrichment analysis was performed against the background of all genes expressed in the dataset. A hypergeometric test was used, and the resulting *p*-values were again adjusted for multiple testing using the BH method. Terms with an adjusted *p*-value (q-value) < 0.05 were considered statistically significantly enriched. The reference gene sets were obtained from the MSigDB (for GO) and KEGG databases as integrated within the clusterProfiler package. Significance of enrichment was defined as false discovery rate (FDR) < 0.05 after Benjamini–Hochberg correction for multiple testing.

### 2.4. Gene Set Enrichment Analysis (GSEA)

GSEA was performed to identify biological pathways significantly enriched in the CFA group compared to the control group [21]. The analysis was conducted using the ClusterProfiler package in R. Genes were ranked based on fold change, and predefined gene sets from the Molecular Signatures Database (MSigDB) database were used as references. Enrichment score (ES), normalized enrichment score (NES), nominal *p*-value, and FDR were used to evaluate the significance of enrichment. Visualization was carried out using the ggplot2 package.

### 2.5. PPI Network Construction and Hub Gene Identification

All identified DEGs were submitted to the STRING database to construct the PPI network (organism: Mus musculus; minimum required interaction score ≥ 0.4) [22]. The resulting network was visualized using Cytoscape software (version 3.8.2), and hub genes were identified using the cytoHubba plugin with the MCC algorithm.

### 2.6. Random Forest Analysis for Gene Ranking

A Random Forest analysis was performed to rank the importance of the DEGs identified in Section 3.1. The analysis was implemented using the randomForestpackage in R. The model was constructed with the following parameters to ensure stability and mitigate overfitting: the number of trees (ntree) was set to 1000. Due to the small sample size, a leave-one-out cross-validation approach was inherently applied by utilizing the out-of-bag (OOB) error estimate. The class imbalance was not an issue as the two groups had an equal number of samples (*n* = 3 per group). The Gini index was used as the criterion for measuring variable importance. The analysis was run 10 times independently to ensure the stability of the gene importance rankings, and the average importance score for each gene was used for the final ranking [23].

### 2.7. Establishment of the CFA-Induced Inflammatory Pain Model and Tissue Collection

Mice were randomly divided into CON and CFA model groups. Complete Freund’s Adjuvant (CFA, Sigma-Aldrich, product number F5881) was prepared at a concentration of 1 mg/mL. The CFA group received a subcutaneous injection of 20 μL CFA into the right hind paw, while the control group received an equal volume of saline. Behavioral observations, including mechanical allodynia and paw swelling, were recorded daily after injection. On day 3 post-injection, mice were anesthetized with sodium pentobarbital (50 mg/kg, intraperitoneally), and the ACC region of the brain was rapidly harvested. Tissue samples were snap-frozen in liquid nitrogen for RNA extraction for histological analysis. For all subsequent behavioral tests, the experimenters performing the measurements were blinded to the group allocation (CFA vs. CON) of the animals. The blinding was maintained until all data collection and analysis were completed. A total of 16 male C57BL/6 mice were used, randomly assigned to the following groups: CFA model group (*n* = 8) and control group (*n* = 8).

### 2.8. Behavioral Tests

Mechanical allodynia, thermal hyperalgesia, and local edema were assessed as described below. All tests were conducted in a quiet room between 13:00 and 17:00 to minimize circadian influences, with the animals acclimated to the testing environment for at least 30 min prior to the start of the experiment.

#### 2.8.1. Assessment of Mechanical Allodynia (Von Frey Test)

Mechanical sensitivity was assessed by measuring the paw withdrawal threshold (PWT) using calibrated Von Frey filaments (North Coast Scientific, Cleveland, OH, USA). Mice were placed individually in clear Plexiglas chambers on an elevated wire mesh grid. After acclimation, a series of von Frey filaments with logarithmically increasing stiffness (ranging from 0.008 to 2.0 g) were applied perpendicularly to the plantar surface of the right hind paw with sufficient force to cause a slight bend, holding for 3–5 s. A sharp withdrawal, shaking, or licking of the paw was considered a positive response. The mechanical threshold was determined using the up-down method as previously described. The testing pattern continued until four measurements were made after the first change in response, and the 50% paw withdrawal threshold was calculated from the resulting pattern.

#### 2.8.2. Assessment of Thermal Hyperalgesia

Thermal hyperalgesia was evaluated using a Hargreaves-type analgesiometer Hargreaves apparatus (IITC Life Science, Woodland Hills, CA, USA) to measure the paw withdrawal latency (PWL). Mice were placed in individual Plexiglas chambers on an elevated glass plate. A mobile radiant heat source was focused onto the plantar surface of the hind paw. The intensity of the heat stimulus was calibrated at the beginning of this study to produce a baseline latency of approximately 10–12 s in naïve control animals and remained constant throughout the experiment. To prevent tissue damage, a maximum cut-off time of 20 s was automatically enforced. The withdrawal latency was measured three times for each paw with at least 5 min intervals between trials, and the average of the three latencies was used for analysis.

#### 2.8.3. Measurement of Paw Edema

Local inflammatory edema was quantified by measuring the thickness of the hind paw using a digital caliper (Mitutoyo, Kanagawa, Japan, accuracy ±0.01 mm). Mice were gently restrained, and the thickness of the right hind paw was measured at the metatarsal level to ensure consistency. The measurement was performed in triplicate for each animal at each time point, and the mean value was recorded as the paw thickness.

### 2.9. Quantitative Real-Time PCR (RT-qPCR)

For qPCR validation, RNA extracted from the ACC region of mice from the behavioral cohorts (CFA group, *n* = 6; Control group, *n* = 6) was used.

#### 2.9.1. RNA Extraction and Reverse Transcription

Total RNA from the ACC was extracted using the Trizol method. RNA concentration and purity (A260/A280 ratio between 1.8 and 2.0) were assessed using a NanoDrop 2000 spectrophotometer (Thermo Fisher Scientific, Waltham, MA, USA), and RNA integrity was verified by 1% agarose gel electrophoresis. Reverse transcription was performed using the HiScript^®^ II Q RT SuperMix for qPCR kit (Fujibio, Shanghai, China), and cDNA was stored at −20 °C.

#### 2.9.2. qPCR Detection

qPCR was carried out using ChamQ Universal SYBR qPCR Master Mix (Fujibio, China) on an ABI QuantStudio 6 Real-Time PCR system (Applied Biosystems, San Francisco, CA, USA). Each gene was tested in triplicate using β-actin as the internal control. The 20 μL reaction mixture included 10 μL of SYBR qPCR Master Mix, 0.4 μL of each primer (10 μM), 1 μL of cDNA, and 8.2 μL of nuclease-free water. The thermal cycling conditions were as follows: 95 °C for 30 s; 40 cycles of 95 °C for 10 s and 60 °C for 30 s. Relative gene expression was calculated using the 2^−ΔΔCt^ method. The primer sequences used in this study are listed in Table 1.

### 2.10. Ethical Statement

All animal experiments were conducted in strict accordance with ethical guidelines for the care and use of laboratory animals, with every effort made to minimize pain and discomfort. The study protocol was approved by the Ethics Committee for Laboratory Animal Care of Chengdu University of Traditional Chinese Medicine (Approval No. 2025011; Date of Approval: 5 February 2025). All procedures complied with the “Guidelines for Ethical Review of Laboratory Animal Welfare” (GB/T 35892-2018) [24] and were carried out in strict accordance with the institutional regulations of Chengdu University of Traditional Chinese Medicine. The official approval document is provided in the Appendix A.

## 3. Results

### 3.1. Dataset Selection and Differential Gene Expression

A total of 76 significantly DEGs were identified, including 64 upregulated and 12 downregulated genes. A heatmap was generated to visualize the expression patterns of these DEGs (Figure 1A–D).

### 3.2. Functional Annotation

#### 3.2.1. GO Enrichment Analysis

GO enrichment analysis was performed to classify DEGs into biological process (BP), cellular component (CC), and molecular function (MF). In the BP category, DEGs were significantly enriched in response to glucocorticoids (GO:0051384, *p* = 3.2 × 10^−5^), cellular response to toxic substances (GO:0097237, *p* = 1.8 × 10^−4^), and lipid transport (GO:0006869, *p* = 4.1 × 10^−3^). In the CC category, DEGs were predominantly localized to collagen-containing extracellular matrix (GO:0062023, *p* = 2.1 × 10^−6^) and plasma membrane rafts (GO:0045121, *p* = 0.002). MF analysis revealed enrichment in lipid transporter activity (GO:0004017, *p* = 0.004) and organic anion transmembrane transporter activity (GO:0015347, *p* = 0.006) (Figure 2A).

#### 3.2.2. GSEA

GSEA was used to further investigate pathway enrichment based on the entire gene expression dataset. Results indicated significant enrichment in cytokine-mediated signaling pathways, inflammatory responses, and cellular responses to cytokines, suggesting the involvement of these processes in CFA-induced chronic pain mechanisms (Figure 2B).

#### 3.2.3. KEGG Pathway Enrichment Analysis

The DEGs were primarily enriched in the TGF-β signaling pathway (ko04350, *p* = 0.008) and regulation of the actin cytoskeleton (ko04810, *p* = 0.012) (Figure 2C).

### 3.3. PPI Network and Machine Learning Analysis

The 76 whole DEGs were used to build the PPI in the STRING database (Figure 3A).

Furthermore, to identify the core genes in the PPI, the cytoHubba plugin was used to rank the genes, and the top 10 hub genes with the highest scores were selected. Among them, *Fn1*, *Bgn*, and *Lum* were identified as potentially critical genes (Figure 3B).

To further prioritize the most influential genes from the DEGs, we employed a Random Forest algorithm. This method is particularly useful for ranking variables by their importance in classifying the groups, rather than for building a generalized predictive model, especially with limited sample sizes. The resulting gene importance scores allowed us to identify a subset of key genes. *Fn1*, *Bgn*, and *Lum* consistently ranked at the top across multiple runs. While the OOB error estimate was low and the AUC reached 1.0, we interpret this primarily as an indication of strong separability within this specific dataset, acknowledging the potential for overfitting. Thus, the primary value of this analysis lies in the robust ranking of gene importance, which converged on ECM-related hub genes (Figure 3C,D).

### 3.4. Experimental Validation of Hub Gene Expression

#### 3.4.1. CFA Model and Behavioral Assessment

The success of the CFA-induced inflammatory pain model was confirmed by evaluating local inflammation and pain hypersensitivity behaviors. We performed three independent behavioral tests, each targeting a specific modality, on separate cohorts of mice to avoid inter-test interference. First, to assess mechanical allodynia, the von Frey filament test was conducted on one cohort of mice (*n* = 8 per group). The mechanical withdrawal threshold of the ipsilateral hind paw was significantly decreased in CFA mice compared to the controls. The threshold dropped from a baseline of 1.46 ± 0.45 g to 0.26 ± 0.18 g on day 1 post-injection and remained at a low level of 0.18 ± 0.09 g on day 3, whereas the threshold in the control group stayed stable at around 1.35 ± 0.54 g (*p* = 0.0009 for days 1 and 3 vs. CON). Subsequently, thermal hyperalgesia was evaluated using the Hargreaves test on a different cohort of animals (n = 8 per group). A pronounced reduction in the paw withdrawal latency to radiant heat was observed in the CFA group. The latency shortened from 13.4 ± 2.1 s at baseline to 8.5 ± 2.3 s on day 1 (*p* = 0.0007), indicating the development of thermal hyperalgesia. The control group latency remained unchanged at approximately 13.5 ± 1.0 s. In addition, paw edema, as a direct measure of local inflammation, was quantified by measuring paw thickness in a third cohort (*n* = 8 per group). A marked increase in paw thickness was evident in the CFA-injected paw. The thickness increased from 1.38 ± 0.20 mm at baseline to 2.15 ± 0.13 mm on day 1 (*p* = 0.001), while the control group showed no significant change (1.36 ± 0.09 mm). Collectively, these results from three distinct behavioral paradigms demonstrate that our CFA model successfully induced robust peripheral inflammation, mechanical allodynia, and thermal hyperalgesia, confirming the establishment of a reliable inflammatory pain model (Figure 4A–D).

#### 3.4.2. qPCR Validation of Gene Expression

Total RNA was extracted and reverse-transcribed into cDNA, and gene expression was assessed by using qPCR. The mRNA expression levels of *Fn1*, *Bgn*, and *Lum* were significantly upregulated in the CFA group compared to the control group (*Fn1*:1.325-fold increase, *p* = 0.02; *Bgn*:1.846-fold increase, *p* = 0.02; *Lum*: 3.142-fold increase, *p* = 0.001) (Figure 4E).

## 4. Discussion

Some DEGs have been found in ACC that suggested the key brain region involved in pain modulation. This indicated that the central sensitization of CFA-induced inflammatory stimuli [25]. Notably, *Fn1*, *Bgn*, and *Lum* exhibited the most prominent fold changes among the DEGs, and they are ECM-associated molecules. Previous studies have shown that ECM remodeling can influence synaptic plasticity by modulating neuron–glia interactions [25]. Therefore, the DEGs identified in this study may participate in central pain regulation through ECM-neuronal interaction networks.

GO and KEGG enrichment analyses revealed that the DEGs were significantly enriched in the TGF-β signaling pathway and ECM organization. The TGF-β pathway plays a dual role in the pathogenesis of RA. On the one hand, it promotes synovial fibroblast activation, contributing to joint destruction [26], On the other hand, it modulates microglial polarization, participating in neuroinflammatory processes [27]. In the present study, TGF-β pathway-related genes in the ACC region—such as Smad3 and Thbs1—were significantly upregulated in the CFA model, suggesting that central inflammatory responses may be driven by activation of this pathway. Furthermore, the upregulation of ECM-related genes such as *Fn1*, *Bgn*, and *Lum* in the ACC may reflect central transcriptional changes corresponding to the excessive ECM accumulation seen in RA synovium, thereby supporting the concept of a regulatory “brain–joint axis” in RA pathogenesis.

Among the core genes identified through both Random Forest analysis and PPI network construction, *Fn1* warrants particular attention. *Fn1* is a central component of the ECM, and previous studies have shown that it can promote the release of proinflammatory cytokines, potentially through activation of the NF-κB signaling pathway [28]. In line with this, our study found that Fn1 expression was significantly upregulated in the ACC. This association raises the possibility that ECM remodeling in the CNS may be involved in the response to peripheral inflammatory challenge such as RA [29]. Additionally, *Lum*, a regulator of ECM stability [30], may contribute to altered pain processing by disrupting the neuronal microenvironment in the ACC when aberrantly expressed Another key gene, *Bgn*, which amplify inflammatory responses through the Toll-like receptor 2/4 (TLR2/4) signaling pathways [31]. Interestingly, our GSEA results demonstrated significant activation of the TLR signaling pathway, suggesting that *Bgn* may serve as a pivotal mediator in this process. It is important to note that our data primarily establish an association between ECM gene upregulation and inflammatory pain. The precise causal mechanisms underlying this association require further investigation through functional studies [32].

This study has several limitations. First, the transcriptomic analysis relied on a dataset with a small sample size (*n* = 3 per group), which may affect the generalizability of the DEGs we identified, despite our experimental validation. Second, the CFA-induced inflammatory pain model, while recapitulating key features of central sensitization relevant to RA pain, does not encompass the systemic autoimmune pathology of the human disease. Therefore, the extrapolation of our findings to RA mechanisms should be considered indicative rather than definitive. Finally, the upregulated expression of *Fn1*, *Bgn*, and *Lum* in the ACC suggests an association with inflammatory pain, but the precise cellular sources and causal roles of these ECM components in central sensitization await future functional investigations.

Future studies should aim to validate these central mechanisms in animal models that more closely recapitulate the systemic autoimmunity of RA. Furthermore, exploring the potential interaction between peripheral factors that modulate systemic inflammation/oxidation and the central ECM-mediated neuroimmune responses identified here represents a fascinating direction for future research [33]. This could provide a more holistic understanding of the disease and uncover novel non-pharmacological intervention strategies.

## 5. Conclusions

This study integrated whole-transcriptome analysis and experimental validation to identify key molecular signatures associated with CFA-induced inflammatory pain. By focusing on ACC, some functions have been enriched such as ECM remodeling and neuroplasticity. *Fn1*, *Bgn*, and *Lum* were prioritized as core genes, and these genes are functionally linked to ECM dynamics and neuroinflammatory processes, suggesting a novel inflammation–ECM–neuroplasticity regulatory axis in chronic pain. These findings provide new molecular insights into the central mechanisms of RA.

## Figures and Tables

**Figure 1 genes-17-00015-f001:**
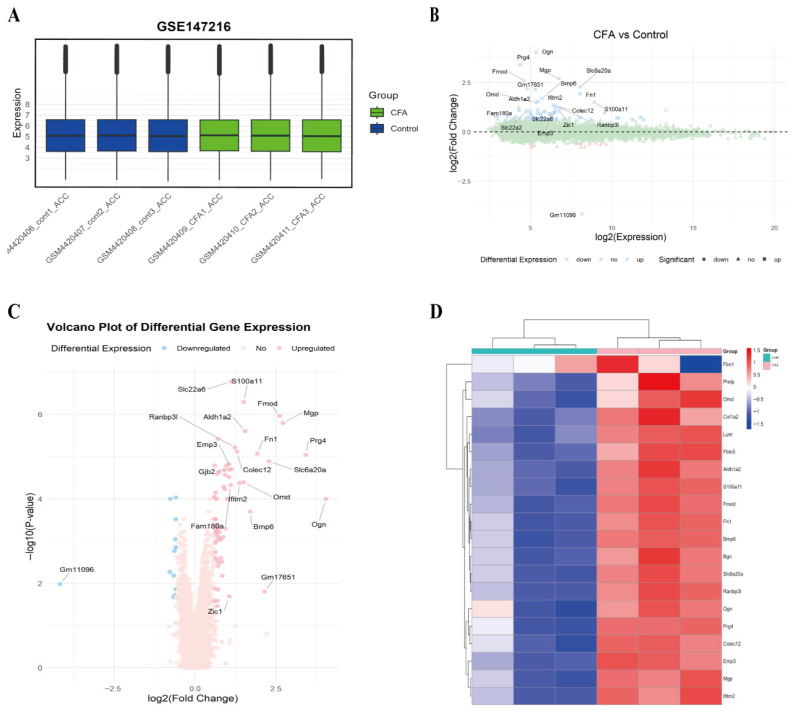
Transcriptomic profiling of the ACC region in the CFA model. (**A**) Boxplot showing the distribution of normalized gene expression values across all samples in the GSE147216 dataset. (**B**) Overview of DEGs identification from the GSE147216 dataset comparing CFA and control samples. (**C**) Volcano plot displaying differentially expressed genes (DEGs). Red dots represent significantly upregulated genes (adjusted *p*-value < 0.05 and log2FC ≥ 0.585), blue dots represent significantly downregulated genes (adjusted *p*-value < 0.05 and log2FC ≤ −0.585), and gray dots represent non-significant genes. (**D**) Heatmap of the top 20 most significant DEGs (rows) across individual samples (columns). Color scale represents Z-score of gene expression. Sample groups are annotated: CFA (*n* = 3) and CON (Control, *n* = 3).

**Figure 2 genes-17-00015-f002:**
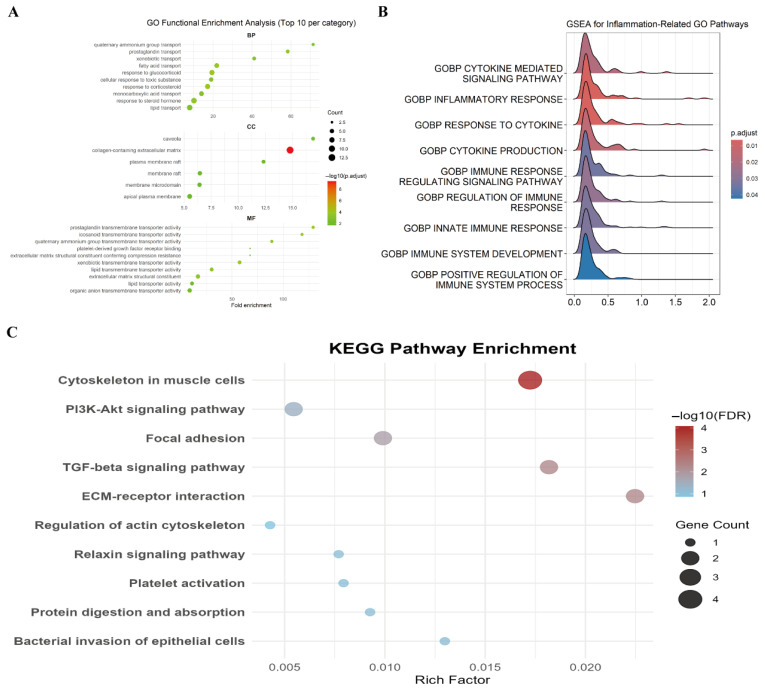
Functional enrichment analysis of DEGs. (**A**) GO enrichment analysis results for BP, CC, and MF categories. The top 10 significantly enriched terms (based on false discovery rate, FDR) per category are shown. (**B**) GSEA for inflammation-related GO pathways. (**C**) KEGG pathway enrichment analysis. The top 10 significantly enriched pathways (FDR < 0.05) are displayed.

**Figure 3 genes-17-00015-f003:**
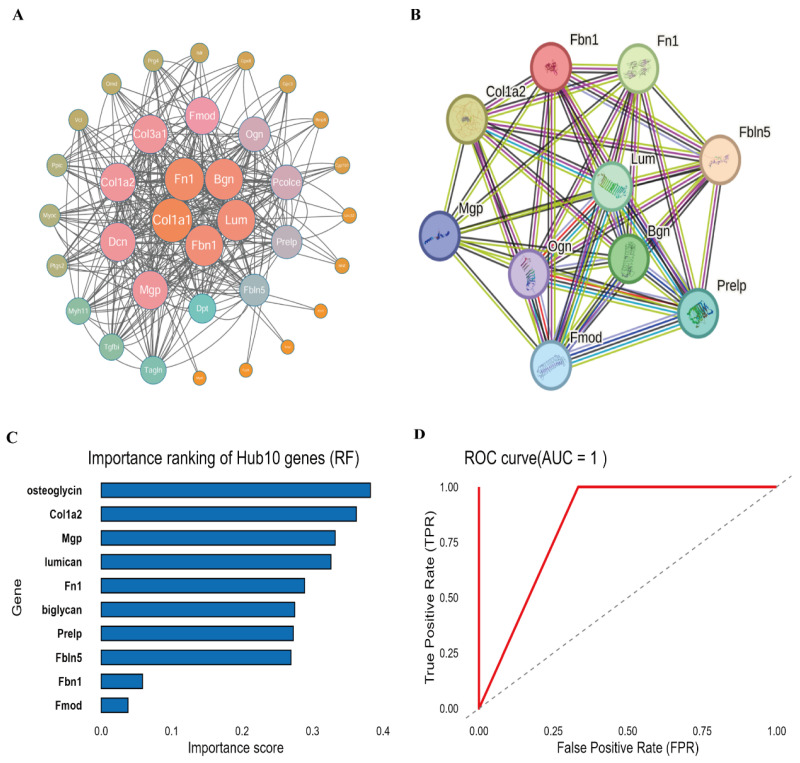
Identification of hub genes by PPI network and machine learning. (**A**) The PPI network of DEGs was constructed using the STRING database (interaction score > 0.4). Nodes represent proteins, edges represent interactions. (**B**) The key subnetwork of hub genes identified by the Maximal Clique Centrality (MCC) algorithm from CytoHubba. (**C**) The importance ranking of the top 10 hub genes as determined by the Random Forest algorithm (mean decrease in Gini index). (**D**) ROC curves demonstrating the diagnostic power of the identified hub genes for classifying CFA and control samples. The AUC for each gene is indicated in the legend.

**Figure 4 genes-17-00015-f004:**
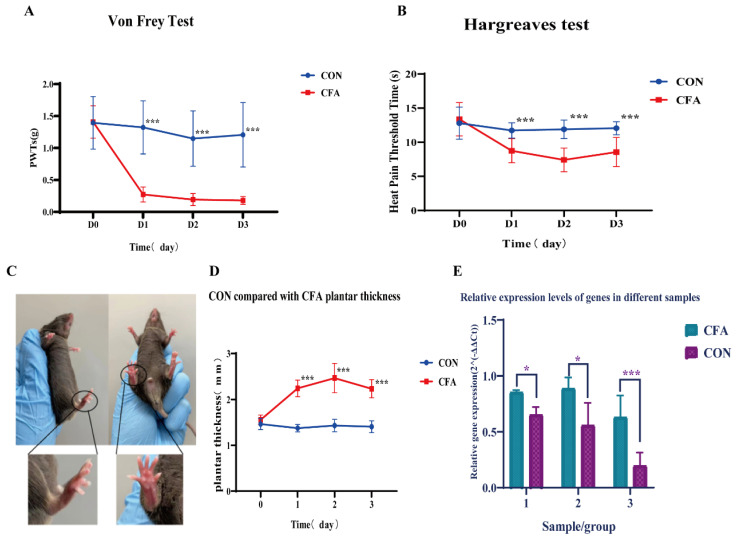
Behavioral results of CFA and mRNA express in hub genes. (**A**) Time course of mechanical pain thresholds assessed by the von Frey filament test in mice injected with CFA (red, *n* = 8) and saline control (blue, n = 8). Data are presented as mean ± SD. *** *p* < 0.001 compared to the control group (two-way ANOVA with Bonferroni’s post hoc test). (**B**) Time course of thermal pain latency assessed by the Hargreaves test in the same cohorts as in (**A**). Data are presented as mean ± SD. *** *p* < 0.001 compared to the control group (two-way ANOVA with Bonferroni’s post hoc test). (**C**) Representative photographs showing paw swelling on day 3 post-injection with CFA (**left**) versus the contralateral saline-injected control paw (**right**). (**D**) Quantitative analysis of paw thickness on day 1 post-injection (*n* = 8 per group). Data are presented as mean ± SD. *** *p* < 0.001 compared to the control group (Student’s *t*-test). (**E**) Relative mRNA expression levels of Fn1, Bgn, and Lum in the ACC region as determined by qPCR (*n* = 6 per group). Data are normalized to the control group and presented as mean ± SD. * *p* < 0.05, *** *p* < 0.001 compared to the control group (Student’s *t*-test).

**Table 1 genes-17-00015-t001:** Primer Sequences Used for RT-qPCR.

Gene	Forward Primers	Reverse Primer
*M-Actb*	CCACCATGTACCCAGGCATT	CAGCTCAGTAACAGTCCGCC
*Fn1*	GCTCAGCAAATCGTGCAGC	CTAGGTAGGTCCGTTCCCACT
*Bgn*	TGCCATGTGTCCTTTCGGTT	CAGGTCTAGCAGTGTGGTGTC
*Lum*	CTCTTGCCTTGGCATTAGTCG	GGGGGCAGTTACATTCTGGTG

## Data Availability

The datasets analyzed during the current study are available in the GEO database under accession number GSE147216 (https://www.ncbi.nlm.nih.gov/geo/query/acc.cgi?acc=GSE147216 (accessed on 15 October 2025)).

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
