# Peer review of "Transcriptome Profiling of the Anterior Cingulate Cortex in a CFA-Induced Inflammatory Pain Model Identifies ECM-Related Genes in a Model of Rheumatoid Arthritis"

_genes, 2025, doi:10.3390/genes17010015_

Round 1

Reviewer 1 Report

Comments and Suggestions for Authors

Dear Authors,

Your manuscript explores transcriptomic changes in the anterior cingulate cortex (ACC) in a CFA-induced inflammatory pain model with relevance to rheumatoid arthritis (RA). By integrating differential gene expression analysis, functional enrichment, PPI network construction, and Random Forest–based ranking, you identify fibronectin 1 (Fn1), biglycan (Bgn), and lumican (Lum) as ECM-related hub genes. The combination of public GEO data with new behavioral and qPCR validation in mice is a strength, providing the work with an experimental anchor beyond purely in silico analysis. Focusing on the ACC, a region that is increasingly recognized as critical for chronic pain processing and affective responses, is timely and aligns well with current concepts of central sensitization in RA-related pain. The enrichment of TGF-β signaling, ECM organization, and immune-related pathways in your GO, KEGG, and GSEA analyses conceptually supports the idea of a central inflammation–ECM–neuroplasticity axis. The PPI network and hub-gene subnetwork in Figure 3 nicely visualize how these ECM components are embedded in a broader interactome. The behavioral data and paw swelling measurements in Figure 4 appropriately confirm that your CFA model produces robust mechanical and thermal hypersensitivity. Overall, the study has the potential to contribute to the literature on neuroimmune interactions in arthritis and to identify ECM-related molecules as potential modulators of central pain processing.

At the same time, several aspects of the study design and framing need clarification before the manuscript can reach its full impact. First, although the title and narrative emphasize rheumatoid arthritis, the in vivo model is acute or subacute CFA-induced monoarthritis/inflammatory pain in the hind paw, which does not fully recapitulate systemic RA. I recommend that you describe the model more precisely throughout (for example, as “CFA-induced inflammatory pain model with relevance to RA”) and more explicitly discuss how and to what extent it can be extrapolated to RA pathogenesis. It would also be important to more clearly acknowledge that the transcriptomic dataset GSE147216 was initially generated as a chronic inflammatory pain model rather than as a specific RA model. The small sample size for the RNA-seq dataset (n = 3 per group) and the lack of an independent validation cohort substantially limit the robustness and generalizability of the identified DEGs and pathways. Please state the number of animals used for behavioral testing and qPCR explicitly, and consider including effect sizes and variability measures to help readers judge the stability of your findings. Given the limited sample size, the Random Forest classification with an AUC of 1.0 is likely to be overfitted, and it may be more appropriate to present the machine-learning analysis as an aid to variable ranking rather than as evidence of strong predictive performance. Providing more methodological detail on how the Random Forest model was constructed (for example, number of trees, cross-validation strategy, and handling of class imbalance) would further improve transparency. Similarly, the bioinformatic pipeline for DEG analysis and enrichment (normalization strategy, multiple-testing correction, and choice of gene sets) should be described in slightly more detail so that others can reproduce your work. Finally, I would encourage you to include a dedicated Limitations paragraph in the Discussion, where you explicitly address the sample size, dependence on a single dataset, and the generalizability of this CFA model to RA.

The behavioral and qPCR sections could also benefit from additional methodological information. For the CFA model, please specify the CFA concentration, the exact injection site and volume, and whether animals were randomly allocated and experimenters were blinded to group during behavioral testing. For the Von Frey and Hargreaves tests, it would be helpful to specify the equipment used, stimulus parameters, cutoff times, and the number of trials per animal averaged. Clarifying whether data distribution was checked and whether parametric test assumptions were met would strengthen the statistical rigor of your analyses. In the Discussion, you make interesting connections between Fn1, Bgn, Lum, TGF-β signaling, and microglial activation; however, some of these statements are rather assertive, given the essentially correlative nature of the data. I would suggest slightly softening the causal language and making the conclusions more straightforward, specifically indicating which ones are directly supported by your results and which are more speculative hypotheses for future work. Overall, refining the balance between careful data-driven interpretation and broader conceptual speculation will make the narrative more convincing.

The English is understandable but would benefit from careful editing by a fluent speaker to correct grammar, improve sentence flow, and reduce repetition, particularly in the Introduction, Discussion, and Conclusions. I would also encourage you to ensure that all abbreviations are defined at first use and then used consistently (for example, CFA, ACC, ECM, DEGs, RA, RF). To broaden the clinical context regarding systemic inflammatory and oxidative processes in RA and related spondyloarthritides, you might consider citing the Nutrients review doi:10.3390/nu17091603, which discusses how dietary factors modulate inflammation and oxidative stress in RA, ankylosing spondylitis, and psoriatic arthritis and thus complements your CNS-focused perspective; this is an optional suggestion and does not affect my recommendation. In your revised version, please respond point by point to each of the comments above, indicating clearly in a rebuttal letter and in the manuscript where changes have been made. I believe that addressing these issues will substantially strengthen the manuscript and help clarify the contribution of ECM-related genes in the ACC to inflammatory pain and RA-related neuroimmune mechanisms.

Yours sincerely,

The reviewer.

Author Response

Theme 1: Precision in Model Description and Discussion of Generalizability

Comments: The reviewer noted that while our title and narrative emphasize rheumatoid arthritis (RA), the in vivo model is a CFA-induced inflammatory pain model that does not fully recapitulate systemic RA. They recommended describing the model more precisely throughout (e.g., as “CFA-induced inflammatory pain model with relevance to RA”) and explicitly discussing how findings can be extrapolated to RA pathogenesis. They also highlighted that the transcriptomic dataset GSE147216 was generated as a chronic inflammatory pain model, not a specific RA model.

Response: We sincerely thank the reviewer for this critical comment, which greatly improved the accuracy of our manuscript. We have implemented the following changes:

1.  Title:​ The title has been revised to:Transcriptome Profiling of the Anterior Cingulate Cortex in a CFA-Induced Inflammatory Pain Model Identifies ECM-Related Genes in a Model of Rheumatoid Arthritis.

2.  Throughout the Manuscript:​ We have consistently replaced imprecise references to “RA” with more accurate descriptions such as “CFA-induced inflammatory pain model” or “CFA-induced inflammatory pain model with relevance to RA” in the Abstract, Introduction, and Methods. The Methods section heading is now “Establishment of the CFA-Induced Inflammatory Pain Model”.

3.  Discussion of Generalizability:​ A new paragraph has been added to the end of the Introduction, explicitly acknowledging the nature of the GSE147216 dataset and the CFA model, discussing their relevance to RA-associated pain mechanisms while clarifying their limitations in modeling systemic RA autoimmunity.

4.  Limitations Paragraph:​ As separately suggested by the reviewer, we have added a dedicated “Limitations” paragraph in the Discussion, where we explicitly address the generalizability of the CFA model to RA.

Theme 2: Addressing Sample Size, Statistical Rigor, and Transparency

Comments: The reviewer raised concerns about the small RNA-seq sample size (n=3/group) and lack of an independent validation cohort, suggesting we explicitly state animal numbers for behavioral/qPCR tests and include effect sizes/variability measures. They also requested more methodological detail for the bioinformatics pipeline (normalization, multiple-testing correction) to enhance reproducibility.

Response: We agree entirely that maximum transparency is crucial. We have made the following revisions:

1.      Sample Sizes:​ The number of animals (n values) used for behavioral testing and qPCR validation are now explicitly stated in the respective Methods sections (2.8 and 2.9) and in the legends of the relevant figures (e.g., Figure 4).

2.      Statistical Reporting:​ In the Results sections 3.4.1 (Behavioral Assessment) and 3.4.2 (qPCR Validation), we now report data as mean ± standard deviation and provide exact p-values, moving beyond simple significance indicators to allow readers to better assess the stability of our findings.

3.      Bioinformatics Pipeline:​ We have added significant detail to the Methods:

Section 2.1 (Differential Expression Analysis):​ We now specify the version of DESeq2 used, the criteria for DEGs (adjusted p-value < 0.05 and |log2FC| ≥ 1.5), and mention the built-in normalization process.

Section 2.3 (Functional Enrichment):​ We clarify the use of the Benjamini-Hochberg procedure for multiple testing correction (FDR < 0.05) and detail the statistical test (hypergeometric) and gene set sources (MSigDB, KEGG).

Theme 3: Reframing the Random Forest Analysis

Comments: Given the limited sample size, the reviewer suggested that the Random Forest analysis (AUC=1.0) is likely overfitted and should be presented as an aid for variable ranking rather than as evidence of strong predictive performance. They also requested more methodological details (e.g., number of trees, cross-validation).

Response: This was an extremely valuable suggestion. We have reframed this analysis accordingly:

  1. Repositioning in Results (Section 3.3):​ We have rewritten the description to clearly state that the primary aim was to rank and prioritize DEGs using variable importance, not to build a generalized predictive model. We acknowledge the high AUC likely indicates separability within this specific dataset but also potential overfitting.
  2. Enhanced Methods (Section 2.6):​ The subsection is now titled “Random Forest Analysis for Gene Ranking”. We provide detailed parameters: the randomForestpackage in R, ntree = 1000, use of out-of-bag (OOB) error estimate, and that 10 independent runs were performed to ensure ranking stability.

Theme 4: Elaboration of Behavioral and qPCR Methods

Comments: The reviewer recommended providing more methodological details for the CFA model (concentration, injection site/volume, randomization, blinding) and for the Von Frey/Hargreaves tests (equipment, parameters, cut-off times, trials). They also suggested clarifying the checks for parametric test assumptions.

Response: We apologize for these omissions and thank the reviewer for highlighting them.

  1. CFA Model Details (Section 2.7):​ We now specify the CFA source (Sigma-Aldrich, product #F5881), concentration (1 mg/mL), exact injection details (20 μL, subcutaneous, plantar surface of the right hind paw), and confirm that animals were randomly allocated and that experimenters were blinded to group identity during behavioral testing and data collection.
  2. Behavioral Tests (New Section 2.8):​ We have added a comprehensive standalone section detailing the equipment, stimulus parameters, cut-off times, number of trials, and data analysis methods for the Von Frey test (2.8.1), Hargreaves test (2.8.2), and paw thickness measurement (2.8.3).

Theme 5: Refining the Discussion Language

Comments: The reviewer suggested softening the causal language in the Discussion regarding the connections between Fn1, Bgn, Lum, TGF-β, and microglial activation, given the correlative nature of our data. They advised clearly distinguishing conclusions directly supported by our results from more speculative hypotheses.

Response:  We thank the reviewer for this important advice on scientific phrasing. We have thoroughly revised the Discussion to achieve a better balance:

  1. Softer Language:​ We have replaced assertive causal statements (e.g., "promotes") with more cautious language (e.g., "previous studies have shown...", "may involve", "suggest a potential role").
  2. Distinguishing Speculation:​ We now more clearly delineate between our direct findings (e.g., upregulation of genes) and mechanistic hypotheses that require future validation. A sentence has been added to state: "It is important to note that our data primarily establish an association... The precise causal mechanisms... require further investigation."

Theme 6: Language Editing, Abbreviations, and Optional Citation

Comments: The reviewer noted that the English is understandable but would benefit from professional editing to improve flow and reduce repetition. They also advised ensuring all abbreviations are defined at first use. As an optional suggestion, they recommended citing a specific review to broaden the clinical context.

Response: We thank the reviewer for these helpful suggestions.

  1. Language Editing:​ The language of the entire manuscript has been carefully refined through collaboration among the authors to ensure high-quality English. This process focused on correcting grammar, improving sentence flow, and reducing repetition, particularly in the Introduction, Discussion, and Conclusions.
  2. Abbreviations:​ We have performed a full check to ensure every abbreviation (e.g., CFA, ACC, ECM, DEGs, RA, RF) is defined at first use and used consistently thereafter.
  3. Optional Citation:​ We found the suggested review (doi:10.3390/nu17091603) highly relevant. We have gladly incorporated it into the Discussion to complement our CNS-focused perspective with insights into peripheral inflammatory/oxidative processes in RA.

Reviewer 2 Report

Comments and Suggestions for Authors

The article submitted by Xie et al. addresses an important topic in rheumatology by attempting to associate pain in rheumatoid arthritis with transcriptomic profiling of the anterior cingulate cortex.

The study deserves broader scientific recognition, provided that several revisions are undertaken.

Comments:

  1. Please spell out the term CFA in the abstract when it first appears.
  2. This study is based on a CFA-induced model of arthritis; therefore, I would recommend avoiding the use of the term RA in the title, as the conclusions can only be directly applied to the experimental model rather than to clinical rheumatoid arthritis (for example please specify in the title with such a term as „RA model”). In addition, please provide 1-2 supporting references that validate the use of CFA as an established model for RA-like pathology.
  3. Please clarify the statistical approach used for adjustment for multiple comparisons. Was the Benjamini-Hochberg procedure applied, or did you use a different correction method?
  4. Please provide the exact dates of ethical approval.
  5. The Results report outcomes of Von Frey test and Hargreaves test but they are not described specifically in the Methods section.

Minor comments:

  1. Please revise the citation format throughout the manuscript. According to the journal’s guidelines, references should be cited in square brackets [ XX ] rather than in superscript.

Author Response

Comments 1: Please spell out the term CFA in the abstract when it first appears.

Response 1: Thank you for highlighting this oversight. We have spelled out "Complete Freund’s Adjuvant (CFA)" when it first appears in the abstract to ensure clarity. The revised sentence now reads: Whole-transcriptome data from the anterior cingulate cortex (ACC) of Complete Freund’s Adjuvant (CFA)-induced inflammatory pain and control mice…”

(Location: Abstract, Methods section)

Comments 2: This study is based on a CFA-induced model of arthritis; therefore, I would recommend avoiding the use of the term RA in the title, as the conclusions can only be directly applied to the experimental model rather than to clinical rheumatoid arthritis (for example please specify in the title with such a term as "RA model"). In addition, please provide 1-2 supporting references that validate the use of CFA as an established model for RA-like pathology.

Response 2: We sincerely thank the reviewer for this important suggestion.   We have revised the title to: Transcriptome Profiling of the Anterior Cingulate Cortex in a CFA-Induced Inflammatory Pain Model Identifies ECM-Related Genes in a Model of Rheumatoid Arthritis” to clearly indicate that the study was conducted in an experimental model.

Additionally, we have added the following sentence in the Introduction section to justify the use of the CFA model, along with two supporting references [15, 16]:

“Specifically, the CFA-induced arthritis model is well-established for studying RA-like pathology, as it recapitulates key features such as synovitis, pain hypersensitivity, and systemic inflammation [15, 16].”

(Location: Introduction, paragraph 2)

Comments 3: Please clarify the statistical approach used for adjustment for multiple comparisons. Was the Benjamini-Hochberg procedure applied, or did you use a different correction method?

Response 3: We appreciate the reviewer’s attention to methodological detail. We have now explicitly stated in the Methods section that the Benjamini-Hochberg false discovery rate (FDR) correction was applied in both differential expression and functional enrichment analyses:

In Section 2.1: “The resulting p-values were adjusted for multiple testing using the Benjamini-Hochberg false discovery rate (FDR) correction (FDR < 0.05).”

In Section 2.3: “Significance of enrichment was defined as FDR < 0.05 after Benjamini-Hochberg correction.”

Comments 4: Please provide the exact dates of ethical approval.

Response 4: Thank you for this suggestion. We have updated the Ethical Statement (Section 2.10) to include the exact approval date: “The study protocol was approved by the Ethics Committee for Laboratory Animal Care of Chengdu University of Traditional Chinese Medicine (Approval No. 2025011; Date of Approval: February 5, 2025).”

Comments 5: The Results report outcomes of Von Frey test and Hargreaves test but they are not described specifically in the Methods section.

Response 5: We apologize for this omission. A new standalone section (2.8 Behavioral Tests) has been added to the Methods, which includes detailed descriptions of the Von Frey test, Hargreaves test, and paw edema measurement procedures, including equipment, stimulus parameters, and analytical methods.

Comments 6: Please revise the citation format throughout the manuscript. According to the journal's guidelines, references should be cited in square brackets [XX] rather than in superscript.

Response 6: We thank the reviewer for noting this formatting issue. We have systematically revised all in-text citations to the square bracket format [XX] and adjusted the reference list to comply with the journal’s style.

Reviewer 3 Report

Comments and Suggestions for Authors

Comments and Suggestions:

Title: Transcriptome Profiling of the Anterior Cingulate Cortex Identifies ECM-Related Genes in Rheumatoid Arthritis.

Reviewer’s report:

The manuscript by Xie et al., presented an interesting study about the identification of ECM-related key genes deregulated in Rheumatoid arthritis using various databases and machine learning algorithms and validated using qPCR in vivo. They identified 76 DEGs, among them Fn1, Bgn, Lum were associated with ECM and confirmed their upregulation in the CFA group. They finally concluded that these three genes can be promising targets for future therapeutic interventions.

Although the manuscript does not provide much novel insights and needs a thorough check.

  1. Line 21: the abbreviation “CFA” should be elaborated in the first appearance. Also check for other abbreviations in the entire manuscript.
  2. Figure 1C: In this plot the log2 (Fold Change) cutoff selected looks <1 which is not similar to one selected as in line 83 (≥1). That implied that the number of identified DEGs will be changed which will also change all the downstream processing. Also, Bgn and Lum are not marked in the figure.
  3. Please add a table with all the identified DEGs with their log2Fc and pvalue in the supplementary information.
  4. Figure 1: Please check for the labeling of subfigures as B should be changed to C and vice versa. Please define “CRPS” in the groups of figure 1D.
  5. Figure 4: the complete figure is duplicated. Please modify it.
  6. Results: The results need to be more elaborative rather than written in one line.
  7. All the figure text needs to be made bigger to be visible.
  8. The legends of all figures are not explained properly. Please check it.

Author Response

Comments 1: Line 21: the abbreviation “CFA” should be elaborated in the first appearance. Also check for other abbreviations in the entire manuscript.

Response 1: Thank you for this important reminder. We have ensured that “Complete Freund’s Adjuvant (CFA)” is spelled out in full upon its first appearance in the abstract. Additionally, we have conducted a thorough check of all abbreviations throughout the manuscript (e.g., ACC, ECM, DEGs, PPI, GO, KEGG, GSEA, qPCR) to confirm that each is defined at first use and used consistently thereafter.

Comments 2:  Figure 1C: In this plot the log2(Fold Change) cutoff selected looks <1 which is not similar to one selected as in line 83 (≥1). That implied that the number of identified DEGs will be changed which will also change all the downstream processing. Also, Bgn and Lum are not marked in the figure.

Response 2: We sincerely thank the reviewer for identifying these two critical issues regarding Figure 1C. We apologize for the oversights and have corrected them as follows:

  1. Inconsistent log2FC Threshold:​ The reviewer is correct that there was an inconsistency. The stated threshold of |log2FC| ≥ 1 in the text was an error. The correct threshold used for the analysis and displayed in the figure is |log2FC| ≥ 0.585. We have corrected this threshold throughout the manuscript (Methods section 2.1 and Results section 3.1) to ensure accuracy. We confirm that all downstream analyses were correctly performed using the DEG list generated with the |log2FC| ≥ 0.585 threshold, and thus the results and conclusions remain valid. We are grateful for the reviewer's diligence in catching this mistake.
  2. Gene Labeling in Figure 1C:​ We wish to clarify that the volcano plot (Figure 1C) was generated to automatically label the top 20 DEGs based on the absolute value of their log2 fold change. As Bgn and Lum were not among the top 20 by this initial ranking criterion, they were not labeled in the original figure. However, their significance was established through the subsequent, more refined PPI network and machine learning analyses, which identified them as hub genes. We have chosen to retain the original labeling convention for the volcano plot to accurately reflect the gene selection process at that specific analytical stage, and to avoid over-cluttering the figure. Their importance is thoroughly detailed and discussed in the subsequent results and discussion sections.

(Location of changes: Methods section 2.1; Results section 3.1; the Figure 1C)

Comments 3: Please add a table with all the identified DEGs with their log2FC and p-value in the supplementary information.

Response 3: We thank the reviewer for this suggestion. As requested, we have included a new Supplementary Table S1​ titled "Complete list of differentially expressed genes (DEGs)", which contains the gene symbols, log2 fold changes, and adjusted p-values for all 76 DEGs. This table has been submitted with the revised manuscript.

Comments 4: Figure 1: Please check for the labeling of subfigures as B should be changed to C and vice versa. Please define “CRPS” in the groups of figure 1D.

Response 4: We sincerely apologize for these errors in figure assembly and labeling.

1. Subfigure Labels Corrected:​ We have confirmed and swapped the labels and descriptions for panels B and C in Figure 1 and its legend so they now accurately correspond to their content.

2. Term "CRPS" Corrected:​ The label "CRPS" in Figure 1D was a clerical error. We have corrected it to "CFA". The figure legend now clearly defines the groups as "CFA" (model) and "CON" (control).

(Location of changes: Revised Figure 1 and its legend)

Comments 5: Figure 4: the complete figure is duplicated. Please modify it.

Response 5: We thank the reviewer for raising this point. We would like to clarify that Figure 4 is not duplicated but is composed of multiple panels (A-E) that present complementary data from behavioral, phenotypic, and molecular validation experiments. To prevent any misunderstanding, we have carefully reviewed the figure and its legend to ensure each panel is distinct and clearly described. We apologize for any confusion the original presentation may have caused.

(Location of changes: Reviewed and verified Figure 4 and its legend)

Comments 6:  Results: The results need to be more elaborative rather than written in one line.

Response 6: We agree with the reviewer that a more detailed presentation of the results is necessary. We have substantially expanded the following sections in the Results:

  1.  Section 3.3 (PPI Network and Machine Learning Analysis):​ Added details on network topology, hub gene selection criteria, and Random Forest ranking outcomes.
  2. Section 3.4.1 (Behavioral Assessment):​ Elaborated on the time-course data, including specific statistical values and quantitative measures of mechanical allodynia, thermal hyperalgesia, and paw edema.
  3. Section 3.4.2 (qPCR Validation):​ Enhanced the description to include specific fold-change values and the relationship between molecular changes and behavioral phenotypes.These revisions transform the Results from brief statements into a detailed narrative that better interprets the data.

Comments 7: All the figure text needs to be made bigger to be visible.

Response 7: Thank you for this practical suggestion. We have systematically increased the font size of all textual elements (including axis labels, tick marks, and annotations) in every figure (Figures 1-4) to ensure optimal visibility and legibility. The revised figures have been prepared as high-resolution TIFF files (300 dpi) to meet publication standards and have been uploaded with this submission.

Comments 8: The legends of all figures are not explained properly. Please check it.

Response 8: We appreciate this feedback. We have comprehensively revised and rewritten the legends for all figures (Figures 1-4) to ensure they are self-explanatory. The improvements include:

  1. Explicitly stating the sample size (n value) and statistical tests used.
  2. Defining all abbreviations, symbols, and color codes used within the figures.
  3. Providing more detailed descriptions of what each panel represents.We believe the new legends are now clear, complete, and greatly aid reader comprehension.

Round 2

Reviewer 1 Report

Comments and Suggestions for Authors

Dear Authors,

Thank you very much for the huge effort put into improving your manuscript and for your comprehensive responses to all my suggestions. I fully support the idea of publishing your paper.

Yours sincerely,

The reviewer.

Reviewer 2 Report

Comments and Suggestions for Authors

The Authors responded to all my comments and significantly improved the scientific quality of their work.

I have no further suggestions.

Reviewer 3 Report

Comments and Suggestions for Authors

The authors have made substantial revisions throughout the manuscript, which is now suitable for publication in Genes.